# Original Research: How Will the TNFD Impact the Health Sector’s Nature-Risks Management?

**DOI:** 10.3390/ijerph192013345

**Published:** 2022-10-16

**Authors:** Tom Deweerdt, Kristin Caltabiano, Paul Dargusch

**Affiliations:** School of Earth and Environmental Sciences, University of Queensland, 1, St. Lucia, Brisbane, QLD 4067, Australia

**Keywords:** TNFD, health sector nature risks, health sector financial risks, health sector risk management, nature opportunities, nature-related financial disclosures, biodiversity loss and financial risks, TCFD, health sector nature dependency, nature risk assessment

## Abstract

The G20-led TNFD taskforce or TNFD is under development and should be ready in September 2023. With one year to go before its official release, the study of the beta versions of the 0.2 framework is crucial to know the strategy of the taskforce regarding metrics and sectors with high natural risks. Its big sister, the Taskforce on Climate-related Financial Disclosure or TCFD, had defined the health sector as a non-key in terms of climate change and carbon risks, but the TNFD decided that it was a priority in terms of nature-related risks and co-dependencies. This research therefore focuses on the innovations of the TNFD and its impact on future disclosures in the sector. The goal being to predict if the TNFD will lead to more disclosures and therefore better risk management from the health firms. To complete this research, the analysis of the sector’s risks and dependencies on nature-related issues and biodiversity loss was essential. To do so, a policy analysis on the framework of the TNFD was conducted, as well as a literature review on nature related risks and opportunities for the health sector. The Health Sector-Wide Approach (SWAp) was analysed to highlight the Task Force’s focus on the health sector. Finally, a due diligence of the TNFD’s stakeholders and partners was carried out to ascertain the interest and participation of health sector actors in the TNFD. Results have shown that Nature risks were important for the sector and that the TNFD was giving more importance to the sector in terms of priority. On the contrary, the health sector does not show an improved interest in this new taskforce. There is a need for more research in the implementation of Nature-financial metrics for disclosures.

## 1. Introduction

The increase in the average temperature of the planet between now and the end of the century will vary according to the region but will be accompanied everywhere by significant changes in the frequency and severity of extreme events [1]. This trend requires companies to integrate climate change and its consequences on natural risks and their activity into their business model [2].

The notion of natural risk covers all the threats that certain natural phenomena and hazards pose to populations, structures, and equipment. More or less violent, these natural events are always likely to be dangerous in human, economic or environmental terms [3]. The prevention of natural risks consists of adapting to these phenomena in order to reduce, as much as possible, their foreseeable consequences and potential damage [4].

The first step towards establishing a risk mitigation strategy is risk awareness [5]. The second step is the anticipation and assessment of these risks [5]. To do so, disclosing according to best practices is essential [6]. Nature risks must be separated from climate change risks in general. The Taskforce on Climate Related Financial Disclosures (TCFD) is developing best practices for climate change risk disclosures but the G20 and G7 have decided that there was a gap in best practices for managing natural risks [7]. To introduce a new international framework for disclosure of these risks, the G20 launched a new Task Force in June 2021 called the TNFD [8]. The TNFD is largely inspired by the TCFD model and framework but develops new standards, targets, and metrics to enable companies to disclose their nature-related financial risks in the best possible way [8]. The TNFD counts already more than 550 members, corporations, assets managers and regulatory agencies from more than 100 countries [8]. Their 6 consultation groups and 16 knowledge partners represent USD 19.4trn in assets [8]. At the moment, the TNFD is in a draft process, a beta version that should lead to an official launch in September 2023 [8]. Nevertheless, this research aims to analyze this new policy and its potential impact on the healthcare sector. Indeed, the health sector is considered non-key for the TCFD, which results, among other reasons, in a low number and quality of climate disclosures by companies in the sector [9]. In contrast, the TNFD considers the health sector to be a priority because it has many risks, opportunities and dependencies related to nature [8]. There is also an important gap in knowledge to be filled on nature metrics and disclosures, the metrics being the cornerstone of disclosure practices for firms, the complexity or lack of them is an obstacle to the popularization of the practice [10]. Comparison with other studies has been a gap in this research as barely any research was conducted on the TNFD Framework before, due to his novelty. The relevance of this research relies on the fact that health care firms are very sensitive to climate risks, yet do not provide sufficient disclosures within institutions [10]. The TCFD and Carbon Disclosure Project (CDP) are key bodies for environmental risk mitigation and demonstrate leadership in this area [6]. The TNFD has similar objectives and may well become the leading body in nature-related risk disclosure. Given that the health sector’s gap in implementing disclosure practice within institutions, it is important to ask: Will the new TNFD Framework lead the health sector to provide more disclosures on nature risk?

To complete this research, the analysis of the sector’s risks and dependencies on nature-related issues and biodiversity loss was essential. To do so, a policy analysis on the framework of the TNFD was conducted, as well as a literature review on nature related risks and opportunities for the health sector. The Health Sector-Wide Approach (SWAp) was analysed to highlight the Task Force’s focus on the health sector. Finally, a due diligence of the TNFD’s stakeholders and partners was carried out to ascertain the interest and participation of health sector actors in the TNFD.

## 2. Materials and Methods

This research is based on a policy analysis [11]. The policy analysis addresses the issue of nature-related risk disclosures within the healthcare sector. The healthcare sector used in this research is defined by the S&P 500, as one of 11 S&P sectors or GICS (Global Industry Classification Standard). It comprises two industry sectors: Healthcare and Equipment’s services and Pharmaceuticals, biotechnology, and services. The policy analysis focuses on the potential answers provided by the beta framework of the Task Force on Nature-related financial disclosures (TNFD). The framework of this new Task Force is analyzed to see if it adequately addresses the nature risk issues of the healthcare sector. To do so, an analysis of the TNFD goals, guidelines, framework, and metrics & targets has been conducted.

To inform this research, a literature review of nature-related risks and opportunities for the health sector was conducted.

The data collection stage involved a search of the research literature using the following parameters. The platforms used were Scopus and Google Scholar, with a date range from 1998 to 2022, and with the following key terms: ‘Nature risks’, ‘Nature opportunities’, ‘Nature-related financial disclosures’, ‘Health Sector Nature risks’, ‘Biodiversity loss and financial risks’, ‘TNFD’, ‘TCFD’, ‘Health sector nature dependency’ and ‘Nature risk assessment’. The research is conducted primarily using peer-reviewed journal articles and international reports. The articles were chosen based on their relevance to the research scope. Most articles were used for the literature review on the health sector relations to nature, on the sector approaches and for the discussion. As only one article mentioned the TNFD, proxy articles on the TCFD were used.

64 articles were identified during the research process. 56 articles were selected for analysis. Certain articles were not selected due to relevancy, date, and scope.

The Health Sector-Wide Approach (SWAp) was analyzed to highlight the Task Force’s focus on the health sector [12].

Finally, a due diligence of the TNFD’s stakeholders and partners was carried out to ascertain the interest and participation of health sector actors in the TNFD. The due diligence looked at the number of participants and stakeholders form the TNFD, related to the health sector [13].

The research design uses: The policy analysis to assess the efficiency of the TNFD framework and if it will lead to more disclosures from the health sector. The literature review on nature risks and opportunities for the health sector, that led the research to find if the sector had many risks from nature, if they were mitigating them and if the TNFD framework was in line with these results. The SWap was used to take a snapshot of the operating principles of the health sector, particularly in ESG practices and the dependency and relation to nature. The due diligence on TNFD stakeholders related to the health sector to assess the interests carried by the sector in the new taskforce.

## 3. Results

The results for the analysis of the TNFD policy draft are based on The TNFD Nature-Related Risk and Opportunity Management and Disclosure Framework Beta v0.2 [8].

### 3.1. Framework

The TNFD framework aims to be a risk management and disclosure tool for financial and non-financial entities to identify and disclose nature-related risks and opportunities. The current version is the second beta of an anticipated total of four, to be released later this year and next. The framework is split into two slightly different versions, to address the differences in assessment and disclosure for financial and non-financial entities but will include identical basic structures: definitions and core concepts, recommended disclosures—including core and sector-specific, targets and metrics, as well as guidelines for incorporating nature considerations into overall risk management.

### 3.2. Goals

The goal of the TNFD is to encourage organizations to endeavor to be more nature-positive, meaning entities contribute positively to nature, rather than purely extractive. The TNFD acknowledges that there is a lack of standard metrics for reporting nature-related risks to investors and the public, and so the information that is currently provided is difficult to interpret and compare across industry. The TNFD aims to fill this gap by developing standard metrics that are consistent with current and anticipated initiatives, such as the SASB and International Sustainability Standards Board (ISSB).

### 3.3. Guidelines

The TNFD framework includes guidelines for both financial and non-financial entities to assist in preparing nature-related risk disclosures, including case studies per sector to assist in preparation. The guidelines comprise of three sections: sector, nature, and realm. The sector level applies to the economic sector that the entity belongs to, as defined by the Sustainable Industry Classification System, in keeping with the TNFD goal of consistency with existing standards. The TNFD has identified eight priority sectors that are expected to be acutely dependent and impacted by nature-related risks and will develop specific guidelines for these sectors, such as specific data requirements for each. Healthcare is one of the priority sectors. Sector-level guidelines aim to maintain equivalencies across classification systems, such as SIC codes in the UK and NAICS codes in the USA. Nature-level guidelines outline dependencies, impacts, risks, and opportunities that apply universally (core concepts) and sector-specifically. It will also provide the business-case for considering nature in strategic decision-making. Realm-level guidelines identify and apply to specific ecosystems, such as ocean, freshwater, terrestrial, and atmospheric.

### 3.4. Targets & Metrics

Proposed targets and metrics are a new feature of the beta v0.2 version, released in June 2022. The targets and metrics are intended to align at three levels of governance:Global, such as dictated by the Global Biodiversity Framework and Sustainable Development Goals.National, such as nation-level regulations and legislation.Local, such as the ecosystem where the business takes place and impacts nature.

The TNFD targets and metrics were designed with six key features:Differentiation between metrics intended for preparing for disclosure (assessment metrics) and those used for actual disclosure (disclosure metrics).Consistency in design to the metrics used for the Task Force for Climate-related Financial Disclosures (TCFD).Applicable to functions up and down the supply or value chain.Core metrics that are intended to be universal, regardless of sector.Evaluated periodically to ensure the recommended metrics are useful and current.Consistency with anticipated international or national frameworks, such as those expected from the Convention on Biological Diversity, that will be negotiated at the upcoming COP-15 later this year, and the Science Based Targets Network.

The TNFD intends to develop core metrics that will be universal, regardless of industry or sector, and flexible to maximize usefulness. These will be consistent across sector, based on relevant regulations, enabling direct comparisons between business entities. Additional disclosure metrics are those that a particular entity decides is useful or relevant to disclose, and the TNFD will provide guidance on choosing relevant metrics and indicators based on their LEAP framework.

The TNFD has developed the LEAP framework to serve as the basis for the metrics that pertain to each sector and has generated two forms of LEAP depending on whether an entity falls under financial or non-financial sectors. It consists of the Locate phase, an assessment metric, assists with determining the current integrity or relative importance of the affected ecosystem. The Evaluate phase, an assessment metric, aids in determining the dependencies and impacts on nature at each location identified (affected ecosystem) in the previous phase. The Assess phase, an assessment metric, includes evaluating the relevant nature-related risks and opportunities that pertain to the entity. Finally, the Prepare phase includes a combination of assessment and disclosure metrics. The assessment metrics assist with deciding on the strategy and resource allocation based on the prior phases, and the disclosure metrics assist in deciding which metrics to disclose, based on the TNFD framework.

### 3.5. Sector Approaches

The pharmaceutical industry represents over 200 countries worldwide [14] representing an over $1 trillion market with growth expected to continue at its fast pace [15]. Historically, health sector ESG reporting has focused on reducing greenhouse gas emissions [14,16], neglecting relevant risk factors such as climate change and its impact on the environment. Climate change’s impact on biodiversity will have negative impacts on human health, such as immunity [17], as well as natural product development presently and in the future. It is estimated that health-related costs associated with climate change will exceed $800 billion per year [17].

The health sector has relied on natural products (NPs) for most of its history—many modern medicines and therapeutics are based on compounds found in nature [18,19], such as Aspirin, Penicillin [18], and Paclitaxel, a common treatment for breast cancer [20]. NPs are compounds and materials sourced from plants, animals, and minerals, etc. [18]. NPs are particularly important for cancer and infectious disease therapeutics, and there is continuing interest in utilizing NPs to address antimicrobial resistance [19]. Further, natural chemical antibiotics are the most effective available [21]. As such, natural products will play an important role in the future of human health and treatment plans and accessibility.

ESG is becoming increasingly important to investors. ESG investments have been increasing year-over-year for the past five years [22], and proper disclosure in ESG reports enables investors to make better decisions as it relates to nature-related risks [23]. Sustainability reporting of S&P 500 companies (not just the health sector) increased from 20% in 2011 to 85% in 2017 [23], with participation likely higher today, however TCFD, and by extension TNFD, is relatively unknown among investors and disclosing firms [23].

Pharmaceutical supply chains lack transparency or disclosure at all [24], making it nearly impossible to assess risks to the business and ongoing viability. A reason for this could be the relative greater significance of human health over environmental health has allowed the health sector to avert scrutiny when it comes to their environmental impact [16]. With the incidence and spread of disease expected to increase as our climate warms [16], ensuring the health sector is contributing in a positive way and can address growing health-related needs will be essential.

While the TNFD provides suggesting reporting metrics, a lack of standardization makes it difficult to compare relative risk across the sector [23]. Firms historically have not been good at measuring physical impacts and consequently financial implications [23]. Information that is disclosed lacks context and so is difficult to interpret and therefore be useful in decision making [23]. To conduct a useful assessment of nature-related risks and opportunities, health sector firms will need to address: raw materials, manufacturing, and packaging. Firms should disclose the source of their raw materials and any risks associated with supply, which would ultimately impact accessibility and price. Manufacturing involves significant water use and waste generation and discharge, and firms should monitor and disclose by-products as part of their operations. Another significant source of materials and waste is packaging of products. Firms should disclose the source of packaging materials and associated waste.

### 3.6. Low Interest from the Health Sector in the Taskforce

The due diligence was carried out on TNFD’s stakeholders, namely:

The TNFD Forum, a consultative grouping of over 600 institutional supporters, the TNFD knowledge partners, a group of 16 leading scientific organizations and standard setting bodies, the TNFD Consultation Groups, informal groups convened in select countries and regions to expand outreach and engagement and the 12 corporate partners and the Stewardship council.

The results show little interest from the health sector in the task force. Indeed, only one of the 12 corporate partners present represents the health sector via the company Glaxo Smith Kline. Of the 600 institutional supporters, only 7 companies were identified, namely Bayer, Royal DSM, Sumitomo Chemical, Nagase & co, Mitsubishi Chemical, GSK, and Ecolab, and only one non-governmental partner: Health in Harmony. This makes a total of 1.3% of institutional supporters. The majority of the other sectors identified as priorities by the TNFD have more stakeholders involved. Within the engagement partners, the Knowledges partners, the consultation groups, and the Stewardship council, none of the stakeholders belong to the health sector.

### 3.7. The Greater Focus of the TNFD on the Health Sector

In the TNFD draft 0.2, the health sector appears twice. Once in the summary and once in the draft, which defines the priority sectors for the task force. The main reason for this paradigm shift is that the pharmaceutical industry relies heavily on Natural Products [19]. In addition, demand for the sector’s products is largely driven by population demographics, rates of insurance coverage, disease profiles and economic conditions [9]. Climate change, biodiversity loss and nature risks are strong pressures to these drivers which make the entire sector under pressure as well [25].

### 3.8. Results on the Health Sector Sensibility to Nature

The health sector depends heavily on nature, most significantly in its supply chain through its use of raw materials for therapeutics as well as packaging as well as in the manufacturing process through water use and waste generation. Besides utilizing NPs for therapeutics, the health sector relies heavily on petroleum-based chemicals in their manufacturing process [25], which also requires a significant amount of water [16]. This necessitates adequate waste management practices [16,17] to minimize among other things drug resistance in organisms in nearby waterways and the environment because of discharge during production [19,26]. The result of these practices has led to the health sector generating more emissions per year than the automotive industry [17].

### 3.9. Nature-Related Risks

Biodiversity is important for drug discover and production; climate change and the loss of biodiversity therefore pose significant threats to the future of drug discovery and production [19,21]. Raw materials, particularly those that are plant-based, are expected to decline with increasing climate change [24]. This is expected to have cascading effects on the entire supply chain, due to the lack of availability and quality of materials that are required for production [27] and therefore have a negative financial impact on firms due to increasing costs and effort required to procure and produce therapeutics and medical devices.

### 3.10. Nature-Related Opportunities

Currently, there is a lack of knowledge around companies’ exposure to nature-related risks and how it will affect profitability and growth [24]. At the same time, consumers are increasingly calling on companies to enhance their environmental stewardship [25,28]. The TNFD provides a framework for the health sector to meet these demands, and standardization will allow for direct comparison between firms [28]. This has the potential for greater accountability among firms in the health sector, as environmentally conscious investors are more likely to pour their money into the firms that perform well on environmental standards and/or have mitigated their exposure to nature-based risks.

Greater accountability can incentivize firms to optimize their operations to more sustainably source materials and manufacture and distribute their products. For the health sector, this involves reducing extraction and waste throughout their operations. From a supply chain perspective, this means reducing raw materials required and therefore extracted and reduced waste generation [17,29]. From a manufacturing perspective, this means less water and energy required for production [17].

## 4. Discussion

The TNFD is expected to bring a non-negligible number of nature-risks disclosures as its framework is modelled after the TCFD. Thus, companies that are already disclosing according to the TCFD should do the same in the TNFD because they are already used to the practice [30]. However, the health sector is behind in practice. Most the world’s leading pharmaceutical and biotechnology companies already disclose within the CDP as a corporate strategy [31].

If these leaders decide to continue the practice within the TCFD, they could influence the market and the sector and bring in other companies [32]. As the practice of disclosure is largely voluntary, the health sector will not benefit from government [33].

In addition, the literature indicates that the introduction of new economic metrics to assess the nature and financial risks will facilitate disclosures for all sectors [34]. The health sector does not have a priority to disclose financial risks related to climate change as it is not a large emitter of greenhouse gases [35,36]. It is different for nature-related risks because the pharmaceutical industry has a high dependence on biodiversity [37]. As the risks are greater, the willingness to analyze and disclose them will also be greater [38]. In contrast, the health industry has a much greater impact on biodiversity than it does on emissions [39]. The sector may face a fear of making public data on its impact on water or on certain species [40]. Indeed, public awareness may result in a financial loss for the pharmaceutical sector [41].

Financial and risk decision-making in firms tend to ignore biodiversity risks and fails to maximize biodiversity conservation [42]. The literature reveals four main principles that are often left out by firms’ management [42,43,44,45,46]. Local and international regulations need to set limits on nature’s exploitation, as finance values certain aspects of nature while other values are considered without value to the investor [42,43,44,45,46]; Nature-related financial risk evaluation needs focus on local and targeted risks due to the nature of biodiversity impacts and dependencies, as seen in this paper [42,43,44]; There is a need to develop biodiversity and nature financing using co-benefits, particularly in conjunction with climate finance [42,43,44,45]; Nature-related finance development needs to be driven by financial decision-makers [42,43,44,45,46].

Nevertheless, the literature argues that the financial sector does not value costs caused by biodiversity loss [42,46]. Financial decision-making cannot value and financially benefit from nature-related investments [42,45]. Additionally, companies generally cannot picture any value concerning risk, impacts and dependency related to nature and biodiversity [42,43,45]. The literature shows that nature-related financial risks cannot be well managed through risk disclosure and quantitative risk estimates because it is simply not efficient enough [45,46]. Instead, many research urge for precautionary policy approaches, which are more binding, and that will lead to a controlled sustainable financial system [45,46]. Nature loss poses risks but are difficult to assign and value for financial decision makers [43,45]. There is therefore a gap in knowledge on research surrounding the new metrics tailored by the TNFD. As the taskforce is in beta version, disclosure related metrics are not finalized yet, and the taskforce does not provide more information on how it plans to overcome the barriers described in the literature.

The pharmaceutical sector has in the past been reluctant to disclose financial and climate information [47]. The literature offers several theories on the reasons behind this reluctance. First, because the private health sector is driven by the intensive use of patents and trade secrets to compete in the creation of vaccines or drugs [48]. Second, the health sector believes it is doing the public good by improving the overall health of its customers and developing life-saving products [35]. Finally, many private sector stakeholders come from the public health sector, or the hospital sector, which is based on professional secrecy and the protection of individual data [49].

To imagine and predict the impact of the TNFD on the increase in the practice of financial disclosures related to nature-risks, it would be necessary to know and be able to measure the impact that the TCFD has had on the increase in quantity and quality of climate disclosures. However, there is a gap in knowledge on this subject, as no scientific study effectively measures the impact of an international organization on private financial practice. Some paradigms emanate from the literature suggesting that the TCFD has indeed helped popularize the practice of climate disclosure [50,51,52], but other studies tend to think that the Taskforce is not effective enough because it is based on the goodwill of companies and stakeholders [53]. Disclosures and carbon accounting are one of the first steps towards environmental risk management for the private sector, but they are also vulnerable to cherry picking and greenwashing [54,55]. The limits attributed to Carbon and Climate disclosures will become limitations for nature-risk disclosures as well [56].

## 5. Conclusions

The policy analysis of the TNFD draft revealed strong similarities with the TCFD. In terms of the framework, goals, guidelines and targets & metrics, some innovations seem to be welcome to enable companies to disclose their nature-related risks. The health approaches sector showed the particularities of the sector regarding natural risks. The sector is intimately linked to nature and biodiversity because it uses natural products to develop its activity. Furthermore, the sector needs transparency in its supply chain and standardization of risk management. The TNFD can therefore meet this need. For the time being, the health sector is not very interested in the TNFD as the due diligence has shown. Indeed, there is little participation of the sector in the taskforce and its bodies. However, the TNFD is very interested in the health sector, considering it a priority because of the risks related to the nature putting pressure on the sector’s stability, as shown by the literature review on risks and opportunities in the health sector. There is a gap in knowledge as to whether the TNFD will lead to an improvement in the quantity and quality of health sector disclosures. If the world’s leading pharmaceutical companies disclose in the TCFD, they should logically do so in the TNFD. This will provide an incentive in the market for competitors. Health companies may be reluctant to disclose on nature because they have a negative impact on nature. Making this impact public could be financially detrimental to companies. Finally, the literature disagrees on the impact of international institutions on increasing disclosures. The TCFD already has strengths and limitations, which bodes well for a similar scenario for the TNFD. This is also supported by the history of the health sector and its traditional reluctance to disclose financial information. A research agenda needs to be set on: 1. How companies can improve the integration of nature into their financial management. 2. The differences between TCFD and TNFD in disclosure practice. 3. The need for new financial metrics related to the loss of value of nature and biodiversity. Further research may be useful before the official release of the TNFD in September 2023. In the health sector, there is also a need for research on the attitudes of the sector and stakeholders towards disclosure practice.

## Data Availability

Not applicable.

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
