# Peer review of "Original Research: How Will the TNFD Impact the Health Sector’s Nature-Risks Management?"

_ijerph, 2022, doi:10.3390/ijerph192013345_

Round 1

Reviewer 1 Report

Dear authors,

The subject is very interesting, but there are some improvements to be made, namely:

-          In the abstract, it is written: “ Taskforce on Nature-related financial disclosure or TNFD”, but I think it would be better as : “Taskforce on Nature-related Financial Disclosure (TNFD).

-          Then the authors write: “Its big sister, the TCFD”, but first must write what is the meaning of TCFD: “Taskforce on Climate related Financial Disclosures (TCFD)

-          The abstract should be more complete, namely with methodology and achieved results.

-          The authors should present more bibliographic references in the text, referring other studies and comparing the results.

-          In line 63, the authors write: “The policy analysis focuses on 62 the potential answers provided by the beta framework of the Task Force on Nature-related 63 financial disclosures (TNFD)”,….but here the authors may write only TNFD.

-          This paper would be considerable enriched if it presents tables with data and figures . As it is , I consider it very poor

-          The authors should present the limitatios and future research.

Author Response

Thank you very much for your recommendations,

Regards

Tom Deweerdt

Reviewer 2 Report

Thank you very much for the opportunity to read the research paper. The following issues are suggested to be considered:

Abstract:

-        In the abstract, the objective of the research, research gap, research method and scope as well as major findings and practical implications should be presented.

Introduction:

-        the scientific research gap, the objective and the aims of the research are not clear.

-        The elements of scientific novelty are not clear. The "Introduction" section shows the current situation, though the scientific significance and what remains unsolved is not revealed. This section should also contain a justification of the need and relevance of the study in the research field.     

-        The structure of the publication is to be explained at the end of the Introduction.

Literature review:

-        The presented paper doesn‘t have a section for literature review and the analysis of scientific literature is not revealed in the paper anywhere. Therefore the current status of the research is not presented.

Methodology:

-        The research design, research ethics, inclusion and exclusion criteria to justify the research scope should be explained. The research method is neither presented nor the process of the research is explained. A methodological explanation is required to understand and prove the choice.

-        The research framework is not presented and justified as an outcome of the scientific literature review.

Discussion:

-        Discussion chapter might be improved by showing a debate between this research results and more recent research publications. This might require also expanding on a literature review so that a research gap would be clearer and based on recent research.

-        The „NFDT“ is used in line 302 without explanation before.

Conclusions:

  • Although the authors present the added value of this paper, the authors might wish to consider explaining the practical and academic implications that should be presented.
  • A future research agenda needs to be provided in the concluding remarks.
  • The conclusions of the research presented should be in the conclusions section.
  • Discussion with the literature review (which is not presented in the research paper)  might better be in the discussion section, e.g. lines 312,313. Also, this kind of sentence should have references referring to the literature sources to justify the sentence. 

Author Response

Thank you very much for your recommendations,

Best regards

Tom Deweerdt

Reviewer 3 Report

The paper presents an interesting topic regarding Task force on nature relate financial disclosers. The language and style of writing is clear and adhere to academic standards.

The abstract and introduction of the paper is weak and needs to be upgraded, first of all the aim of the paper is not clear and needs to be better explained.

Methodological framework is based on a policy analysis with a general description of the search results. what is the main reason for choosing 64 articles? on the basis of which parameters were articles chosen? It is unclear how the results were achieved. Is there any methodological process?

Also Conclusion needs to be upgraded. It is highly recommended to make a stronger connection with results and the goals of the paper. I suggest the author(s) work on these issues and submit a new more detailed version of the paper to the journal when ready.

Author Response

(The authors gave the same response as above.)

Round 2

Reviewer 1 Report

Thank you for all the improvements

Reviewer 2 Report

Even if there is some room for additional improvement left, the authors have made substantial amendments to address the comments provided.